# Photon upconversion with directed emission

K. Börjesson[1,2], P. Rudquist[3], V. Gray[2] & K. Moth-Poulsen[2]

Photon upconversion has the potential to increase the efficiency of single bandgap solar cells beyond the Shockley Queisser limit. Efficient light management is an important point in this context. Here we demonstrate that the direction of upconverted emission can be controlled in a reversible way, by embedding anthracene derivatives together with palladium porphyrin in a liquid crystalline matrix. The system is employed in a triplet-triplet annihilation photon upconversion scheme demonstrating controlled switching of directional anti Stokes emission. Using this approach an emission ratio of 0.37 between the axial and longitudinal emission directions and a directivity of 1.52 is achieved, reasonably close to the theoretical maximal value of 2 obtained from a perfectly oriented sample. The system can be switched for multiple cycles without any visible degradation and the speed of switching is only limited by the intrinsic rate of alignment of the liquid crystalline matrix.

[1] Department of Chemistry and Molecular Biology, University of Gothenburg, Kemigården 4, 412 96 Gothenburg, Sweden. [2] Department of Chemistry and Chemical Engineering, Chalmers University of Technology, Kemivägen 10, 412 96 Gothenburg, Sweden. [3] Department of Microtechnology and Nanoscience, MC2, Chalmers University of Technology, Kemivägen 9, 412 96 Gothenburg, Sweden. Correspondence and requests for materials should be addressed to K.B. (email: karl.borjesson@gu.se).

Photon upconversion, the process of combining low energy photons into high energy photons is interesting, both from a fundamental and applied perspective. Possible applications includes light emitting devices, solar cells and biological imaging. One possible mechanism for photon upconversion is through triplet-triplet annihilation photon upconversion (TTA-UC) in molecular species. TTA was first observed in anthracene solution more than 50 years ago[1], and soon thereafter the concept of combining a sensitizer with TTA as to produce upconverted light was presented[2]. During recent years, a rapid expansion in research efforts is seen, with the explicit goal to use photon upconversion to enable the use of sub-bandgap photons in single bandgap solar harvesting devices[3]. One possible advantage of sensitized triplet-triplet annihilation as compared to other upconversion techniques is that non-coherent, low intensity light is able to drive the process. In fact, as high external quantum efficiencies as 38% (10% under sunlight conditions) has recently been demonstrated[4].

The TTA-UC process involves two different molecular species; a sensitizer (Sen) and an annihilator (An). The process occurs through a series of events, here described briefly (Fig. 1): first, the sensitizer absorbs a photon and due to a fast rate of inter system crossing, the initially formed singlet state is rapidly converted to the relatively long-lived triplet state (Equation 1)[5,6].

$$^1Sen + h\upsilon_1 \rightarrow {}^1Sen^* \rightarrow {}^3Sen^* \tag{1}$$

$$^3Sen^* + {}^1An \rightarrow {}^1Sen + {}^3An^* \tag{2}$$

$$2 {}^3An^* \rightarrow {}^1An + {}^{1,3,5}An^* \tag{3}$$

$$^1An^* \rightarrow {}^1An + h\upsilon_2 \tag{4}$$

The second molecule, the annihilator (An), collides with the excited sensitizer and triplet-triplet energy transfer from the sensitizer to the annihilator occurs (Equation 2). Later, two annihilators in their excited triplet state collide and triplet-triplet annihilation occurs, leading to one annihilator relaxed to the ground state and the other excited to an energetically higher singlet, triplet or quintet state (Equation 3). It was initially believed that spin statistics would limit the probability of the formation of the singlet state to 1/9 (ref. 7), which is the desired state from where emission of a photon occurs (Equation 4). However, examples of yields greater than 1/9 has disproven this

hypothesis[4]. Importantly, the frequency of the emitted photon is higher than that of the initially absorbed photons leading to upconversion of the absorbed light.

Today, several molecular systems have been used as annihilators, including anthracene and derivatives thereof[8–19], perylene[18–26], BODIPY[26–28] and rubrene[29–31] derivatives. The requirement of a fast intersystem crossing of the sensitizer has made metalloporphyrins a popular choice[13–18,23–27,30–33], however, also ruthenium complexes such as Ru(dmb)$_3$ and Ru(bpy)$_3$ derivatives[10] have been used as well as some metal free sensitizers such as C$_{70}$ (ref. 19) and BODIPY chromophores[20–22]. The main motivation for this research field is based on the prospect of driving a chemical or physical reaction, such as a solar cell, utilizing sub-bandgap photons[5]. Proof-of-principle devices demonstrating triplet-triplet annihilation upconversion facilitated water splitting[13], and energy storage via photoisomerizations[34] have been reported. Also a number of photovoltaic cells have been integrated together with upconverting systems[35–37].

In all examples of upconversion, the emission occurs from isotropically oriented molecules, which results in light emitted at equal intensity in all directions. For an application point of view, the upconverted emission should preferably be directional in order to utilize as much light as possible. To achieve directional upconversion, the annihilator molecules needs to be oriented in space. Matrixes that are able to orient molecules in space include stretched films and liquid crystals. Anthracene has previously been shown to orient in accordance to its physical aspect ratio when dissolved in a liquid crystalline matrix having a nematic phase[38,39]. Also, anthracene derivatives having intrinsic liquid crystalline properties have been developed[40,41].

Here we present a method of dynamically regulating the direction of the upconverted emission. It is based on the well-known anthracene/palladium porphyrin upconversion system dissolved in an orientationally ordered liquid crystalline matrix. The orientation factor in the liquid crystalline matrix is controlled by introducing subtle changes in the anthracene substitution pattern. In the bi-component solution only the anthracene derivatives exhibits orientational order. The porphyrin sensitizer is orientationally disordered and therefore absorbs photons from all directions. Using this approach an emission ratio of 0.37 between the axial and longitudinal emission directions and a directivity of 1.52 is achieved, reasonably close to the theoretical maximal value of 2 obtained from a perfectly oriented sample.

## Results

**Orientation of molecules dissolved in liquid crystals**. For an oriented molecule the intensity of emission has a cos$^2$ dependence between the normal of the transition dipole moment and the emission direction[42]. For non-oriented ensemble measurements, this angle dependence is averaged out, leading to uniform emission in all directions. When comparing the emission intensity between an oriented and non-oriented sample, the largest differences occur orthogonal and parallel to the transition dipole moment (Fig. 2a,b). Orthogonal to the transition dipole moment the emission intensity of an oriented sample is twice that of a randomly oriented sample. This is then the maximal gain of orienting molecules for application purposes where the main goal is to drive a secondary process as efficiently as possible. For switching purposes, where the aim is to modulate the upconverted emission by changing the orientation of the molecules, the difference in emission intensities parallel and normal to the transition dipole moment must be compared. Since the emission probability parallel to the

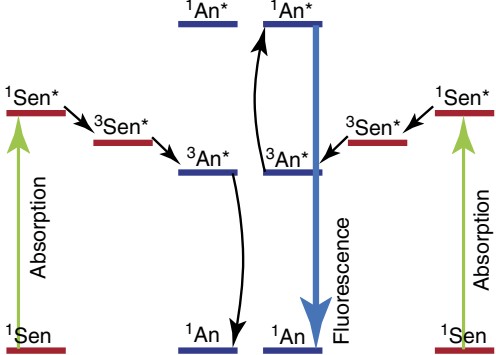

**Figure 1 | The process of photon upconversion.** A sensitizer (Sen) absorbs a photon, which energy after an intramolecular intersystem crossing and an intermolecular triplet energy transfer event, ends up in the excited triplet state of the annihilator (An). When two annihilators in their triplet state meet triplet-triplet annihilation occurs, exciting one of the annihilators to its excited singlet state simultaneously relaxing the other annihilator to its ground state. Importantly, the photon emitted from the excited singlet state has a higher frequency than the initial absorbed photons.

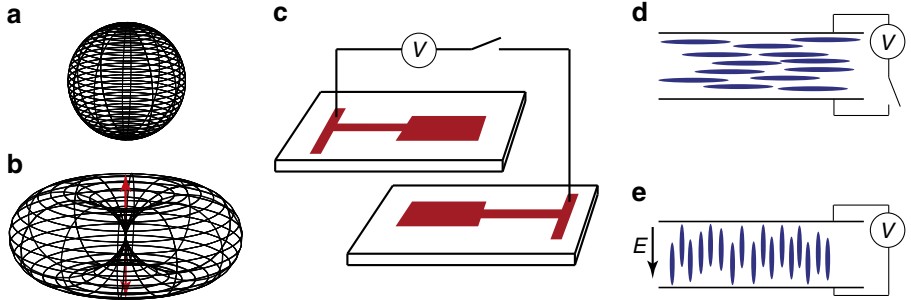

**Figure 2 | The angular emission intensity of an oriented molecule and a liquid crystal cell.** Emission intensity as a function of angle for a random oriented sample (**a**) and for an oriented sample (**b**) with its transition dipole moments in the direction of the red arrow. Schematic illustration of the cell used (**c**). The LC-cell consists of two glass slides patterned with indium tin oxide (ITO) electrodes. The two slides with the electrodes in a face-to-face arrangement are separated using 9 or 45 μm spacers, and on the electrodes conventional rubbed polyimide aligning layers provide a uniform, in-plane orientation direction for the nematic phase in absence of an electric field. Schematic illustration of the molecular orientation of E7 in absence (**d**) and presence (**e**) of an applied voltage.

transition dipole moment direction is zero, the theoretical upconverted signal for a perfectly oriented sample can be modulated between zero and the maximal value.

To orient the annihilators, a liquid crystal (LC) matrix together with technology adopted from the field of liquid crystal displays was employed. As LC matrix the classic mixture E7 was used which provides a nematic phase at room temperature (Supplementary Fig. 1)[43]. The upconversion system dissolved in E7 was through capillary forces inserted into an LC-cell which was subsequently sealed with a solvent resistant epoxy glue, under inert atmosphere. The LC-cell consists of two glass slides patterned with indium tin oxide (ITO) electrodes (Fig. 2c). The two slides with the electrodes in a face-to-face arrangement are separated using 9 μm or 45 μm spacers, and on the electrodes conventional rubbed polyimide aligning layers provide a uniform, in-plane orientation direction for the nematic phase in absence of an electric field (Fig. 2d). In presence of an applied external electric field (**E**, Fig. 2e) the director (the local average of the molecular axes) reorients under the influence from the dielectric torque into the vertical field direction with a characteristic response time, $\tau_{on} \propto E^{-2}$. When the field is switched off, the LC relaxes back to the quiescent uniform state due to elastic torques with a characteristic relaxation time, $\tau_{off} \propto d^2$, where $d$ is the LC layer thickness. For the relatively large values of $d$ used in this study, $\tau_{off}$ is typically much larger than $\tau_{on}$.

Linear dichroism measurements were performed in order to quantify how well the individual components orient in the liquid crystalline matrix (Figs 3, **1–4**). The reduced linear dichroism ($LD^R$) value of a chromophore, with parallel direction of the transition dipole moment with regards to the molecular long axis, is given by the orientational order parameter $S$:

$$S = \frac{1}{2}\langle 3\cos^2\beta - 1 \rangle \tag{5}$$

where $\beta$ is the angle between single molecules and the director **n**. $S$ takes the values 0 for an isotropic system, 1 for perfect orientational order along **n**, and $-0.5$, for $\beta = 90°$ (ref. 44). Starting with the porphyrin sensitizer, no or low orientation of the transition dipole moments can be observed (Fig. 3). This is an important result because the random orientation will allow the sensitizer to absorb light from all directions and with all polarizations, which is a desirable feature in the light absorbing part of the system. However, it is not a surprising result since the sensitizer has no obvious long axis direction which makes physical orientation in the matrix less likely. Moreover, the transition dipole moment is located in a plane and not along a

specific direction which gives a low intrinsic sensitivity on polarized absorption even if oriented.

Three different 9-10 substituted anthracene derivatives (Fig 3a, **1–3**) were explored as annihilator species. The anthracenes were chosen to have the same core chromophore structure but bearing different physical aspect ratios. Furthermore, one of the annihilators, 9-(4-cyanophenyl),10-phenylanthracene (**1**), contains a polar cyano group just like the components in the liquid crystalline matrix. The cyano group could contribute to the dielectric anisotropy of the system and influence the switching dynamics. It might also play a role in the orientational order of **1** in E7 due to dipole correlation effects of the cyano groups in the system. Parallel and antiparallel dipolar correlation within E7 itself, has e.g. been discussed and identified by molecular dynamics simulations[45]. For **1** and 9,10-diphenylanthracene (**2**) the $LD^R$ value is 0.5 and 0.4 respectively, which is slightly lower than the orientational order parameter $S = 0.58$ for the E7-matrix[46]. The fact that the dyes are less ordered than the host is the common situation but there are systems where the dye exhibits larger orientational order than the nematic host[47]. The transition dipole moments for the anthracene molecules are directed along the 9,10 position of the anthracene unit, which means that the molecules orient in this direction in the liquid crystalline matrix, since the $LD^R$ values are positive. Furthermore, the $LD^R$ value of **1** is higher compared to **2**, which can be rationalized with the slightly larger long axis of **1** as compared to **2**. The third annihilator used in this study, 9,10-dimethylanthracene (**3**) shows a low and slightly negative $LD^R$ value. This is consistent with a low orientational order with the average anthracene direction being along the LC director.

**Directional upconverted emission**. Diphenylanthracene (**2**) is probably the most thoroughly studied annihilator in the field of TTA-UC, having reported upconversion quantum efficiencies of 1, 3.2, 7.7, 12.5, and 18% (calculated so that the theoretical maximal efficiency is 50%)[13,15,16,48,49]. **1** and **3** have also been used as annihilators in TTA upconversion applications, with reported quantum efficiencies of 7.9 and 4.6%, respectively[48,50]. All three annihilators are able to upconvert light in the liquid crystalline matrix. The upconversion quantum yields for **1**, **2** and **3** dissolved in E7 in a LC-cell were determined to 0.76, 0.58, and 0.43, respectively, and the transition from the quadratic to linear regime occurs roughly at an excitation power of 400 mW cm$^{-2}$ (Supplementary Fig. 2). To compare these values, the upconversion quantum yield of **1** dissolved in toluene, prepared in the same glovebox was determined to 1.4% using the

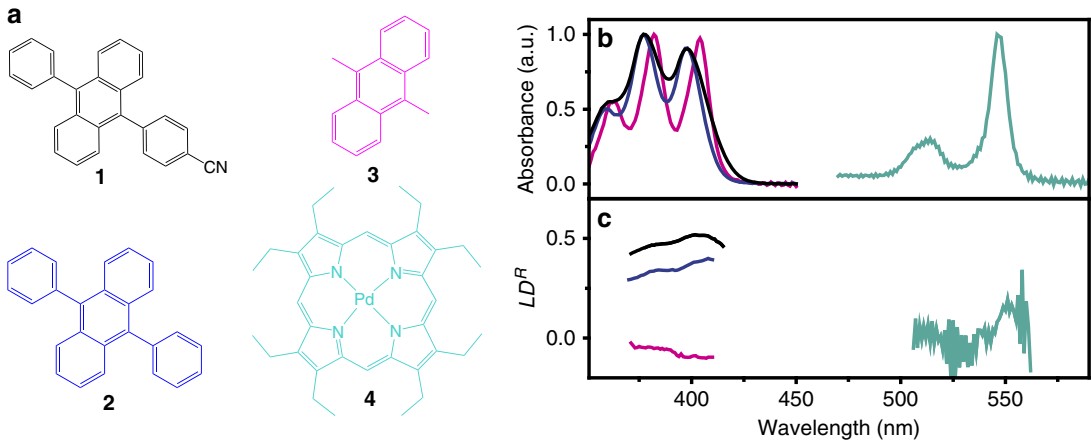

**Figure 3 | Structure of the molecules used in this study, and their envelope of absorption and reduced linear dichroism.** (**a**): 9-(4-cyanophenyl), 10-phenylanthracene (**1**), 9,10-diphenylanthracene (**2**), 9,10-dimethylanthracene (**3**), and palladium(II)octaethylporphyrin (**4**). Isotropic absorption (**b**) and reduced linear dichroism (**c**) of **1** (black), **2** (blue), **3** (magenta), and **4** (turquoise) when dissolved in the liquid crystal matrix E7.

same excitation source. This was done in order to cancel effects of sample preparation (the quantum yield is oxygen sensitive, and thus dependent on method of preparation), and for any difference between excitation source (for example pulsed versus cw lasers). The relative small difference in upconversion quantum yield between samples prepared in the same manner indicate that the liquid crystalline phase as such does not seem to severely hamper the efficiency of the upconversion. The viscosity of the E7 matrix is 40 cps (compared to 0.59 cps for toluene)[51], which explains the higher transition power between the quadratic and linear regime as compared to literature values done in toluene. The higher viscosity of E7 influences the molecular diffusion, which in turn affects the bimolecular quenching constant, $k_q$, for the energy transfer reaction between the sensitizer and annihilator. In toluene the bimolecular quenching constant for the energy transfer reaction between **4** and **1** is $1.85 \cdot 10^9 \ M^{-1} \ s^{-1}$ (ref. 48), and in E7 it is $1.55 \cdot 10^7 \ M^{-1} \ s^{-1}$ (Supplementary Fig. 3). The bimolecular quenching constant is inversely proportional to the viscosity of the media, thus, the difference in the bimolecular quenching constant between samples prepared in toluene and E7 can within a factor of two be explained by differences in the viscosity of the two solvents.

To determine the upconversion switching ratio, emission at 430 nm orthogonal to the surface was monitored at 0 and 14 $V_{rms}$ across the LC-cell (Fig. 4). At 14 $V_{rms}$ the response of the cell is essentially saturated (that is the LC-matrix is fully oriented orthogonal to the surface, Supplementary Fig. 4). Multiple switches back and forth can be done without sign of degradation. For all annihilators, the emission is strongest when no voltage is applied, which indicates that the transition dipole moments of the electronically excited molecules are oriented along the LC director in the nematic phase. **1** displays the largest switching, with an emission ratio of 0.37 in presence versus in absence of the applied electric field. The ratio of **2** and **3** is 0.51 and 0.87, respectively. The emission ratios of **1** and **2** can be rationalized in light of the $LD^R$ data, both orienting with the transition dipole moments along the director in the nematic phase and **1** showing the largest orientation factor of the two. The results of the measurements on molecule **3** are however a little bit surprising, the $LD^R$ data indicate a slight orientation of the anthracene core along the nematic phase, whereas the upconversion switching indicates a slight orientation in the perpendicular direction. This indicates that the absorption and emission data cannot be directly compared since one probes the orientation of ground state

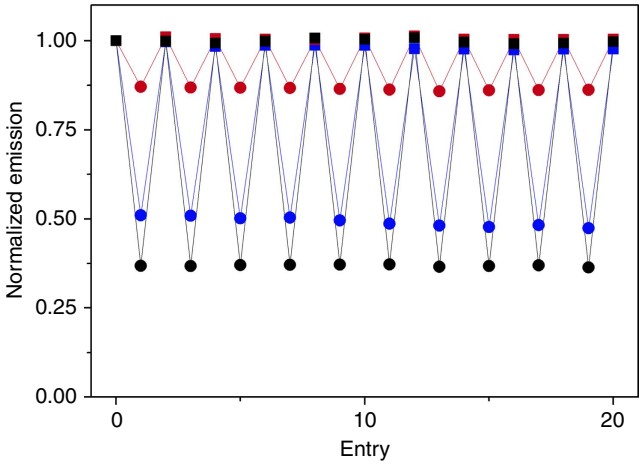

**Figure 4 | Switching of upconverted emission.** Normalized upconverted emission of **1** (black), **2** (blue), and **3** (red) in presence (circles) and absence (squares) of an applied electric field of 14$V_{rms}$. The 10 full switching cycles show negligible degradation and high reproducibility between cycles. Excitation wavelength was 547 nm and the emission was monitored at 430 nm. Concentration of annihilators was 1 mM and of sensitizer 100 μM.

molecules and the other the orientation of excited state molecules in which the electron density is slightly shifted.

To gain a deeper understanding of the switching process and importantly prove that switching does not occur from any trivial effect, such as changes in matrix viscosity, the combination of **1** together with **4** was further studied. Here it is important to bear in mind that in this study the excitation intensities are relatively low which means that the upconverted signal depends quadratically on the emission intensity (Supplementary Fig. 5) and in this regime the upconversion efficiency is viscosity dependent (diffusion controlled). Figure 5 show upconverted emission monitored through polarizers of cells with matrix alignment layers in horizontal or vertical position. As evident from Fig. 5, the largest signal occurs when alignment and polarizers are parallel and at 0 $V_{rms}$. Worth to note is that the signal strength is more or less the same for cases with an applied electric field, and when the emission polarizer is orthogonal to

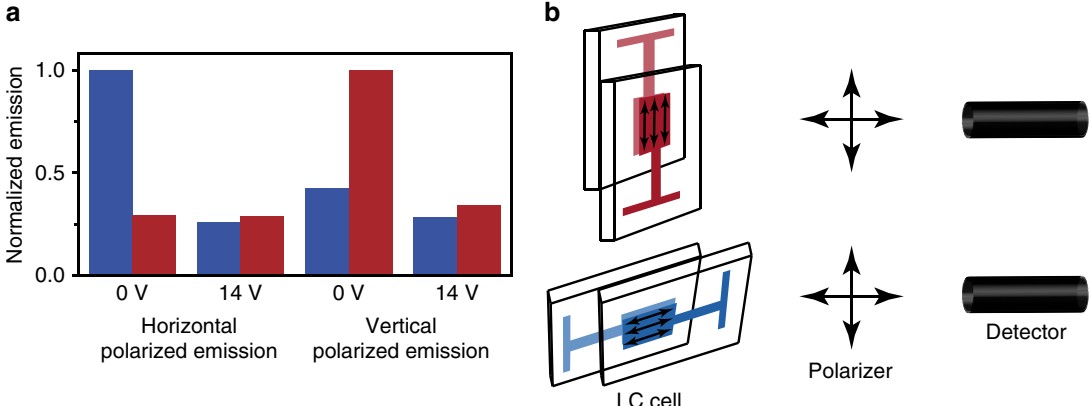

**Figure 5 | Polarized upconverted emission. (a)**, Upconverted emission from an LC-cell with the LC matrix in a horizontal (blue) or vertical (red) position, at 0 or 14 $V_{rms}$, monitored through a polarizer in horizontal or vertical position. Concentration of **1** and **4** was 1 mM and 100 µM, respectively. Excitation wavelength was 547 nm and the emission was monitored at 430 nm. **(b)**, Schematic illustration of the setup of the experiment with the LC-cell either in a vertical (red) or horizontal (blue) position, and the emitted light monitored either through a vertically or horizontally positioned polarizer.

the alignment direction. This is expected due to symmetry of the system, and it proves that the switching of upconverted emission is due to alignment of molecules, and not viscosity changes in the LC matrix when applying an electric field.

The polarized emission data from Fig. 5 was used to assess the increased efficiency of the system, that is the increase in emission orthogonal to the surface for oriented compared to randomly oriented molecules. This allows for determination of the system efficiency increase in an accurate way, since factors affecting the efficiency of the upconversion process such as temperature, viscosity and solvent, is left unchanged. It is assumed that the emission mathematically can be explained by one perfectly ($S = 1$) and one randomly ($S = 0$) oriented population. In Fig. 5 all emission from the cases with an applied electric field as well as with alignment layer and polarizer in orthogonal positions is purely resulting from the randomly oriented population. For the two cases with parallel alignment layer and polarizer the emission comes both from the perfectly and randomly oriented populations. The fraction of perfectly oriented molecules was calculated to be 0.52 by assuming that the fraction of randomly oriented molecules is the same in all cases (the electric field does not increase the orientational order but only reorients the system) and that the emission strength from the oriented population is twice that of the randomly oriented population in the direction perpendicular to the LC director, polarized along the LC director. Based on the fraction of perfectly oriented molecules, the directivity of the system was determined to be 1.52 which is reasonably close to the theoretical maximum value of 2 (see Supplementary Note 1 for details).

So far, only emission normal to the surface of the LC-cell has been considered. For the studied systems, emission normal to the surface is at its maximal value in absence of applied electric field, and emission planar to the surface is at its maximal value in presence of an applied field. This gives rises to an angle dependence of the ratio of the upconverted light in presence and absence of an electric field (Supplementary Fig. 6). Depending on application, either the emitted light normal or in plane to the surface can be utilized. In the former the illumination can be simply made through the front glass with collection of photons through the rear glass. Additionally, if for example the solar cell is transparent for sub-bandgap photons and placed on top of the upconverter, both illumination and collection can be done through the front glass[34–37]. However, for a general high efficiency system, a more sophisticated arrangement is necessary. Regarding the in plane geometry, the emitted light can be confined through total internal reflection (waveguide geometry) and travel to a photon collector at the edges of the device. This is a geometry that already is in use in solar concentrators[52,53]. As the distance from the up conversion event to the detector then can be several millimetres or even centimetres, self-absorption losses must be considered, that is up-converted photons are again absorbed by the system itself before reaching the detector. Self-absorption losses are mainly caused by the absorption by the sensitizer. For the experimental parameters used in this study, roughly half of the emitted photons are reabsorbed by the sensitizer after traveling 1 cm (Supplementary Fig. 7). To minimized losses in the in plane geometry, the absorption wavelength range of the sensitizer should be separated from the emission range of the annihilator. This will require rational design of both sensitizers and annihilators in combination with the chosen liquid crystal matrix.

**Rate of switching**. For applications where switching is the desired feature, such as for example in liquid crystal displays, the rate of the switching process is of great importance. Figure 6a,b shows upconverted emission normal to the cell plane versus time with the electric field over the LC-cell being turned on and off. As evident from Fig. 6a,b, the rate of equilibrating the system differs substantially when the electric field is turned on compared to when it is turned off. When turning on the electric field the emission switches from the high emissive state to the low emissive state within 10 milliseconds. The reversed process, when the field is switched off is slower, taking about 200 milliseconds. Figure 6c also shows the electro-optic response of the LC matrix, displaying switching on the same timescale as for the upconverted emission. This shows that it is the LC-induced order and LC-controlled direction of the molecule that solely determines the performance of the system.

**Discussion**

In summary, we show here that it is possible to achieve photon upconversion having emission in a predefined direction. This without compromising on the systems performance on absorbing photons from all directions in order to drive the process. Furthermore, we show that it is possible to switch the direction of the upconverted light and that the speed of switching is determined by that of the liquid crystalline matrix. The generality of the approach is displayed by the fact that the physical shape of the molecules seems to be the most essential system property and

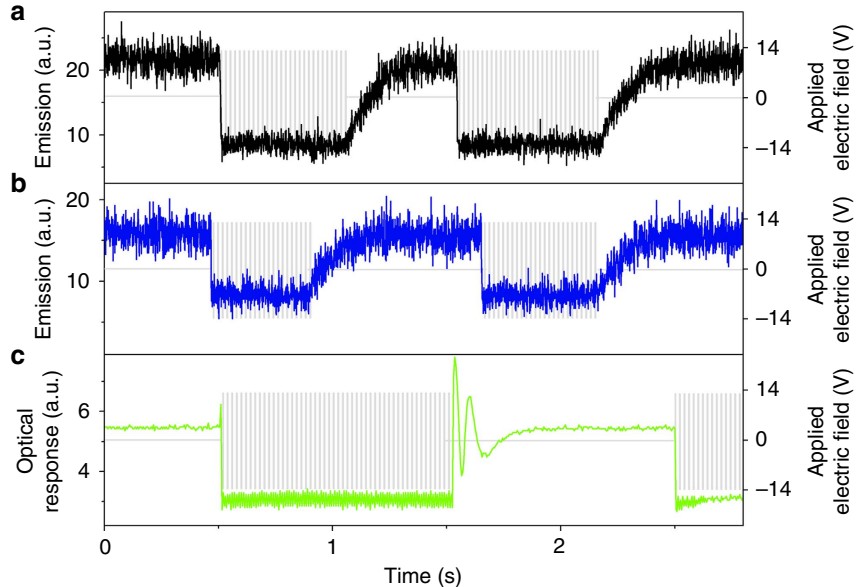

**Figure 6 | Rate of switching. (a)** and (**b**), Upconverted emission (at 430 nm) as a function of time for the annihilators **1** (black) and **2** (blue) with the AC-electric field (grey) being put on and off stepwise. (**c**), Electro-optic switching of pristine E7 cell between crossed polarizers, aligned at 45° from the polarizers, with the AC-electric field (grey) being put on and off stepwise. The transmitted intensity varies with the director orientation out of the cell plane and the dynamics of the director switching is identical to the dynamics of the upconverted emission signals. The oscillations in the electro-optic response originates from the optical retardation of the cell going through several wavelengths under the reorientation of the LC director. The dynamics of the emission and electro-optic switching are the same showing that the emission switching is a result of the field-controlled orientation of the LC.

that most organic molecules can be chemically modified to have a long axis in a desired direction. The concepts and results herein is of fundamental importance when upconversion is used to drive a secondary process, such as a solar cell, as efficient as possible. We have demonstrated a proof of concept using standard dyes as well as a standard liquid crystal host material. As in all guest-host liquid crystal systems the choice of both the dye and the liquid crystal host, as well as their mutual compatibility, are crucial for optimal efficiency of any device based on the directed up-conversion principle described.

## Methods

**General remarks.** All commercially available chemicals were used as received. 9,10-dimethylanthracene was a kind gift of late Prof. Hans-Dieter Becker and purity was confirmed by $H^1$-NMR spectroscopy prior to use. 9-(4-cyanophenyl),10-phenylanthracene was synthesized according to literature procedure[48].

**LC cell manufacturing.** Liquid crystal cells were manufactured in-house at the Chalmers Nanofabrication Laboratory, MC$^2$. Indium tin oxide (ITO) electrodes were created on ITO-sputtered 1.1 mm thick soda-lime glass by means of optical lithography and wet etching. The alignment layer of polyimide PI2610 (DuPont) was spin-coated, thermally cured and subsequently unidirectionally buffed with a velvet cloth in a commercial buffing machine (LCTec Automation) to ensure alignment of the LC director at the surfaces. The plates were cemented together separated with spacer spheres of 9 μm or 45 μm to assure a well-defined LC layer thickness. The buffing directions on the two plates were set parallel to ensure a uniform director field. All processing, including cell assembly was done on 3 in. x 3 in. substrates, which were subsequently cut into 25 identical sample cells. Electric leads were attached to the electrodes by means of ultrasonic soldering. The cells were filled with LC material by means of capillary forces with the LC-porphyrin-anthracene mixtures. Electro-optic response of pristine E7 was measured using a computer controlled arbitrary waveform generator (WFG600, FLC Electronics), in a polarized light microscope (Leitz Orthoplan) with a high-speed photodetector (FLC Electronics) attached to the microscope tube. The electro-optic signal, *i.e.* field-controlled transmission when the cell was placed between crossed polarizers was recorded with an oscilloscope (Agilent technologies, DSO7034A).

**Spectroscopic characterization.** Upconverted emission was measured on a SPEX fluorolog 3 spectrofluorimeter (JY Horiba) equipped with Glan polarizers in the

emission beam when needed. The excitation wavelength was in all cases 547 nm. The linear dichroism (LD) measurements were performed using a Varian Cary 5000 spectrophotometer equipped with Glan air-space calcite polarizers in both sample and reference beam. LD is defined as:

$$LD(\lambda) = A_{\parallel}(\lambda) - A_{\perp}(\lambda) \qquad (6)$$

where $A_{\mathrm{II}}(\lambda)$ and $A_{\perp}(\lambda)$ are the absorption (A) of light polarized parallel (II) and perpendicular ($\perp$) to the macroscopic orientation axis (direction of the nematic phase), respectively. The reduced linear dichroism ($LD^R$) of a uniaxial sample is defined as:

$$LD^R(\lambda) = \frac{A_{\parallel}(\lambda) - A_{\perp}(\lambda)}{A_{\parallel}(\lambda) + 2A_{\perp}(\lambda)} \qquad (7)$$

**Upconversion Quantum Yield measurements.** Upconversion quantum yield measurements were performed on the above mentioned SPEX fluorolog equipped with a 532 nm laserpointer ($I_{max} = 33.8$ mW, spotsize = 0.0573 cm$^2$) as the light source and a graduated neutral density filter was used to vary the intensity of the excitation light. The upconversion quantum yield was measured relative to Cresyl Violet and determined according to:

$$\phi_{UC} = \frac{\phi_r \left( A_r F_x I_r n_r^2 \right)}{A_x F_r I_x n_x^2} \qquad (8)$$

where the subscripts denote the reference (r) and sample (x) respectively and $\phi$ is the quantum yield, A the absorption, F the integrated emission, I the excitation intensity and n the refractive index. For the matrix E7 the refractive index is 1.80 and 1.54 for light parallel and perpendicular to the matrix directors respectively, therefor the quantum yield is reported in the interval of the two extremes[54].

**Data availability.** The authors declare that the data supporting the findings of this study are available from the corresponding author upon request.

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

## Acknowledgements

Prof Bo Albinsson is acknowledged for fruitful discussions and for providing access to spectrophotometer and spectrofluorometer. The authors acknowledge funding from the Swedish Energy Agency (36436-1), Chalmers Materials Area of Advance, the Swedish Research Council (B0361301), Swedish Strategic Research Council (ICA14-0018 & FFL-5) and Knut and Alice Wallenberg foundation.

## Author contributions

K.B., P.R. and K.M-P. designed the experiments. K.B, V.G. and P.R. conducted the experiments. All authors contributed to analysing the results and in writing the manuscript.

## Additional information

**Competing financial interests:** The authors declare no competing financial interests.

