## [Peer Review File · Nature Communications]

Reviewers' comments:

Reviewer #1 (Remarks to the Author):

The authors have responded thoughtfully and in detail to the previous review, especially on the topic of vertical vs horizontal. The strength of the paper is its originality, demonstration of switching, and the relative measurement of vertical vs horizontal emission. I concur with Ref #2, however, that a measurement of quantum yield would benefit the paper. Certainly, I think readers will be wondering about the efficiency in these matrices, and may be frustrated if they cannot find the number. As it is, the wide range of efficiencies reported in prior work serve only to emphasize the need to measure the efficiency in this structure.

Some kind of quantification of efficiency aside, I support publication.

Reviewer #2 (Remarks to the Author):

The authors have made changes to their manuscript in order to address the concerns of the reviewers in their original submission. This reviewer read the reference brought up by Reviewer 1 (Mulder et al) where a liquid crystalline host was used to orient dye molecules for luminescent solar concentrators. Comparing this submission to Mulder et al, the latter seems much more rigorous, with absolute quantum yield measurements combined with models of the optical path, etc.

The main issue I have is that this work introduces the concept of directed emission using liquid crystals at the expense of the upconversion quantum yield (QY). While the liquid crystals can clearly orient the molecules involved in triplet-triplet annihilation (TTA)-based up conversion (this by itself is not new), their presence inhibits efficient triplet energy transfer between the sensitizer and annihilator, resulting in an overall system that operates poorly in the quadratic regime (how poorly we do not know because the authors cannot measure the QY). It appears that this is not a solution to the problem. Perhaps the annihilators could be directly incorporated as part of the liquid crystal? There is no mention of varying the amount of E7 liquid crystal matrix vs. the sensitizer/ annihilator, even though this must have been optimized.

For a proof of concept, this paper does the job, but it does not address the issues that arise regarding the poor upconversion QYs.

Response to Reviewer Comments

Reviewer #1 (Remarks to the Author):

The authors have responded thoughtfully and in detail to the previous review, especially on the

topic of vertical vs horizontal. The strength of the paper is its originality, demonstration of switching, and the relative measurement of vertical vs horizontal emission. I concur with Ref #2, however, that a measurement of quantum yield would benefit the paper. Certainly, I think readers will be wondering about the efficiency in these matrices, and may be frustrated if they cannot find the number. As it is, the wide range of efficiencies reported in prior work serve only to emphasize the need to measure the efficiency in this structure.

Some kind of quantification of efficiency aside, I support publication.

We have now determined the upconversion quantum yield in the liquid crystal matrix. We further determined the upconversion quantum yield for the same molecules in toluene (the standard solvent) prepared in the same way as for the liquid crystalline samples (same glovebox and same cw laser excitation source) in order to give as accurate comparison between the two solvents as possible. The difference between in the QY in toluene and in the liquid crystalline matrix was less than a factor 2.

Reviewer #2 (Remarks to the Author):

The authors have made changes to their manuscript in order to address the concerns of the reviewers in their original submission. This reviewer read the reference brought up by Reviewer 1 (Mulder et al) where a liquid crystalline host was used to orient dye molecules for luminescent solar concentrators. Comparing this submission to Mulder et al, the latter seems much more rigorous, with absolute quantum yield measurements combined with models of the optical path, etc.

We believe that our study is quite rigorous, especially after this last revision in which we have added a quite large set of new experiments.

The main issue I have is that this work introduces the concept of directed emission using liquid crystals at the expense of the upconversion quantum yield (QY).

We have now done the QY measurements and do not see any significant “expense of the upconversion quantum yield”.

While the liquid crystals can clearly orient the molecules involved in triplet-triplet annihilation (TTA)-based up conversion (this by itself is not new), their presence inhibits efficient triplet energy transfer between the sensitizer and annihilator, resulting in an overall system that operates poorly in the quadratic regime (how poorly we do not know because the authors cannot measure the QY).

We have now done a so called Stern-Volmer analysis of our system, in which the triplet energy transfer efficiency between the sensitizer and annihilator can be assessed. We observe a triplet energy transfer that approximately is limited to the rate of diffusion. The energy transfer process is thus very efficient. However, the viscosity of the E7 matrix is somewhat higher than in toluene which gives a lower value of the bimolecular quenching constant.

Furthermore, we have determined at which point our system switches between the so called quadratic to the linear regime, and determined the quantum yield (QY) of our system in the linear

regime. We did this using a cw laser, and compared the QY value of our liquid crystalline system with a conventional system prepared and analysed in the same way (same glovebox for sample preparation and same laser for excitation).

It appears that this is not a solution to the problem. Perhaps the annihilators could be directly incorporated as part of the liquid crystal? There is no mention of varying the amount of E7 liquid crystal matrix vs. the sensitizer/ annihilator, even though this must have been optimized.

We have determined the QY at two different sensitizer concentrations, to show on the effect of the sensitizer concentration on the QY.

For a proof of concept, this paper does the job, but it does not address the issues that arise regarding the poor upconversion QYs.

We do not agree that the upconversion quantum yield is "poor", however before this last revision we did not have any experimental claims for saying this.

Reviewers' comments:

Reviewer #1 (Remarks to the Author):

I think the manuscript is better for the addition of the efficiency measurement. I support accepting the manuscript for publication.

Reviewer #2 (Remarks to the Author):

The authors have fully characterized their system by measuring the up conversion QY and the power dependence. This paper can be accepted without further revision.